Significant changes in the skin microbiome mediated by the sport of roller derby

Meadow James F. 1 jfmeadow@gmail.com
Bateman Ashley C. 1
Herkert Keith M. 1 2
O’Connor Timothy K. 1 3
Green Jessica L. 1 4
1 Biology and the Built Environment Center, Institute of Ecology and Evolution, University of Oregon , Eugene, OR , USA
2 Oregon Health & Science University , Portland, OR , USA
3 Department of Ecology and Evolutionary Biology, University of Arizona , Tucson, AZ , USA
4 Santa Fe Institute , Santa Fe, NM , USA
Cohan Frederick
Electronic publication date: 2013 Mar 12
Publication date: 2013
Volume: 1
Electronic Location ID: e53
Received 2012 Nov 14; Accepted 2013 Feb 28
Copyright: © 2013 Meadow et al.
Copyright year: 2013
Copyright holder: Meadow et al.
License: This is an open access article distributed under the terms of the Creative Commons Attribution License, which permits unrestricted use, distribution, and reproduction in any medium, provided the original author and source are credited.
License URL: https://creativecommons.org/licenses/by/3.0/

Keywords: Microbial biogeography, Contact sport, Human microbiome, Microbial ecology, Skin microbiology, Microbial dispersal

Funding: Alfred P. Sloan Foundation award number 2010-5-22 IEC University of Oregon Research reported in this publication was supported by the Alfred P. Sloan Foundation under award number 2010-5-22 IEC and the University of Oregon. The funders had no role in study design, data collection and analysis, decision to publish, or preparation of the manuscript.

==============================
Diverse bacterial communities live on and in human skin. These complex communities vary by skin location on the body, over time, between individuals, and between geographic regions. Culture-based studies have shown that human to human and human to surface contact mediates the dispersal of pathogens, yet little is currently known about the drivers of bacterial community assembly patterns on human skin. We hypothesized that participation in a sport involving skin to skin contact would result in detectable shifts in skin bacterial community composition. We conducted a study during a flat track roller derby tournament, and found that teammates shared distinct skin microbial communities before and after playing against another team, but that opposing teams’ bacterial communities converged during the course of a roller derby bout. Our results are consistent with the hypothesis that the human skin microbiome shifts in composition during activities involving human to human contact, and that contact sports provide an ideal setting in which to evaluate dispersal of microorganisms between people.

Introduction

Microbial communities living on and in the human skin are diverse and complex. These communities, which vary greatly both within and among people, play an important role in human health and well-being. Skin microbial communities have been shown to mediate skin disorders, provide protection from pathogens, and regulate our immune system (Costello et al., 2009; Grice & Segre, 2011; Human Microbiome Project Consortium, 2012). Despite the importance of our skin microbiota, we still know very little about what shapes the distribution and diversity of the skin microbiome.

As for any other ecosystem, the composition of the skin microbiome is determined by some combination of two simultaneous ecological processes: the selection of certain microbial species by the skin environment and the dispersal of microbes from a pool of available species. Skin moisture, temperature, pH and exposure to ultraviolet light are all well documented environmental factors that affect skin microbial communities (Grice & Segre, 2011). The microbial species available for dispersal onto the skin of any given individual likely stem from many sources including inanimate surfaces, people, pets, cosmetics, air and water (Capone et al., 2011; Costello et al., 2009; Dominguez-Bello et al., 2010; Fierer et al., 2008; Fujimura et al., 2010; Grice & Segre, 2011; Hospodsky et al., 2012; Human Microbiome Project Consortium, 2012; Kembel et al., 2012). Our current understanding of the relative contributions from these potential sources is nascent. Human to surface and human to human contact have long been acknowledged as strong vectors for microbial dispersal in the medical literature, which has been largely focused on culture-based detection of single-species pathogen transmissions (Boyce et al., 1997; Casewell & Phillips, 1977; Hamburger, 1947; Noble et al., 1976; Pessoa-Silva et al., 2004; Pittet et al., 2006). In these culture-based studies, handshaking, as well as hand-contact with other parts of the body and room surfaces, have been identified as strong vectors of health care service infections, such as with methicillin-resistant Staphylococcus aureus (MRSA) and Klebsiella spp. (Casewell & Phillips, 1977; Davis et al., 2012; Pittet et al., 2006). Given that human contact with surfaces, and especially the skin surfaces of others, has been shown to transfer individual microbial taxa, activities which involve human to human contact could be hypothesized to result in the sharing of skin microbial communities.

Here we explore how activities involving human to human contact influence the skin microbiome. We use a contact sport, flat track roller derby, as a model study system. Flat track roller derby is an organized team sport, played worldwide, that involves individuals roller-skating in close proximity and making frequent contact with other players. Roller derby teams frequently engage in tournaments, where teams from different geographical locations come together to play, or ‘bout’ against one another for several days at a time. Flat track roller derby tournaments present an ideal setting in which to study the transmission of skin microbial communities during a contact sport for two main reasons. First, they provide an opportunity to assess if the skin microbiome from athletes that frequently come into contact with one another – members of the same team – have similar microbiomes. Second, they provide an opportunity to assess if skin microbiomes of athletes on opposing teams become more similar after competing against one another. Specifically, we addressed the following questions in our study: (1) Were players’ skin microbiomes predicted by team membership; (2) Were team-specific skin microbiomes altered during a bout; and (3) Did opposing teams’ skin microbiomes become more similar, or converge, after competing in a bout?

Materials and methods

Flat-track roller derby

For a full explanation of approved Women’s Flat Track Derby Association rules, refer to www.wftda.com. Briefly, a bout consists of two 30-min periods, where two competing teams, each composed of up to 4 “blockers” and 1 “jammer”, circle a track with the goal of facilitating their own jammer in accumulating points. Players, both blockers and jammers, periodically rotate with players on the bench, so that few or no players actually play for the entire 60 min bout. Points are accrued when one team’s jammer makes her second, and subsequent, pass through the pack of blockers, in effect lapping the pack. Activity occurs in intervals called “jams”, and a single jam lasts for a maximum of 2 min. Flat track roller derby is a contact sport; blockers are allowed to initiate contact with another player to compete for track position using any of the following body parts: upper arm (shoulder to elbow), torso, hips, “booty” (official WFTDA nomenclature), and mid to upper thigh. Roller derby tournaments often involve multiple pairwise bouts in a single day between several teams, one home team and multiple visiting teams from different geographical locations. Players within a team practice together on a regular basis, and thus come into frequent physical contact, and live in or near the same city. Teams involved in this study were from Eugene, OR (Emerald City Roller Girls); Washington, DC (DC Roller Girls) and San Jose, CA (Silicon Valley Roller Girls).

Ethics statement

Written consent forms were signed and collected from all participating subjects. The Institutional Review Board Initial Application Form for the study was reviewed and approved by the University of Oregon IRB with the Office for Protection of Human Subjects in January 2012 (protocol #10262011.038). The Willamalane Park and Recreation District Human Resources office granted written permission for the study to take place in their recreation facility. Written permission was acquired from the three teams’ coaches and administrators.

Sample collection

Microbial communities inhabiting skin vary greatly across the human body (Grice et al., 2008; Grice et al., 2009; Human Microbiome Project Consortium, 2012). We chose the upper arm (approximately the distal end of the lateral deltoid) as our focal skin sample site. The upper arm is the one skin region on roller derby skaters that is nearly universally exposed and frequently contacted during a bout. All players sampled had this area exposed during the entire bout. All samples were collected Feb. 10, 2012, at the “Big O” Tournament in Eugene, OR, USA, and all biological samples were taken by trained technicians using sterile technique. The two bouts that were sampled took place at 12:00pm (Emerald City vs. Silicon Valley) and 6:00pm (Emerald City vs. DC). DC had already played in one bout the same day at 10:00am (against Santa Cruz, not considered here), but Emerald City and Silicon Valley had not played that day prior to bout 1. For the purposes of this study, players were not monitored between bouts. Samples were collected by swabbing individual’s upper arms in a c. 4 cm by 5 cm area of skin with nylon-flocked swabs (COPAN Flock Technologies, Brescia, Italy) moistened with sterile buffer solution (0.15M NaCl, 0.1M Tween20). Both arms were swabbed on each player at each sampling point, and all samples were taken within 30 min of the beginning and end of each bout. Samples were stored at −20 °C until DNA extraction. Total number of jams was recorded for each player, and multiplied by 2 min (maximum jam length) to approximate total time played per person. Four swab samples were also taken from the floor of the facility (track) following the tournament using the same swabbing method and surface area as the arm samples.

DNA Extraction, amplification and sequencing

Whole genomic DNA was extracted using the MO BIO PowerWater DNA Isolation Kit (MO BIO Laboratories, Carlsbad, CA) according to manufacturers instructions with the following modifications: swab tips were incubated with Solution PW1 in a 65 °C water bath for 15 min prior to bead beating; bead beating length was extended to 10 min since swab tips were included; and samples were eluted in 50 µL Solution PW6. Dual-arm samples from each player were combined for DNA extraction.

A fragment of the 16S rRNA gene including the V4 region was amplified using a modified F515/R806 primer combination (5′-GTGCCAGCMGCCGCGGTAA-3′, 5′-TACNVGGGTATCTAATCC-3′) (Caporaso et al., 2011b; Claesson et al., 2010). Amplification proceeded in two steps using a custom Illumina preparation protocol, where PCR1 was performed with forward primers that contained partial unique barcodes and partial Illumina adapters. The remaining ends of the Illumina adapters were attached during PCR2, and barcodes were recombined in silico using paired-end reads. Adapter sequences are detailed in Supplemental Data. All extracted samples were amplified in triplicate for PCR1 and triplicates were pooled before PCR2. PCR1 (25 µL total volume per reaction) consisted of the following ingredients: 5 µL GC buffer (Thermo Fisher Scientific, U.S.A.), 0.5 µL dNTPs (10 mM, Invitrogen), 0.25 µL Phusion Hotstart II polymerase (Thermo Fisher Scientific, U.S.A.), 13.25 µL certified nucleic-acid free water, 0.5 µL forward primer, 0.5 µL reverse primer, and 5 µL template DNA. The PCR1 conditions were as follows: initial denaturation for 2 min at 98 °C; 22 cycles of 20 s at 98 °C, 30 s at 50 °C and 20 s at 72 °C; and 72 °C for 2 min for final extension. After PCR1, the triplicate reactions were pooled and cleaned with the QIAGEN Minelute PCR Purification Kit according to the manufacturers protocol (QIAGEN, Germantown, MD). Ten µL of 3M NaOAc (pH 5.2) was added to decrease the pH of the pooled reactions and facilitate efficient binding to the spin column during cleanup. Samples were eluted in 11.5 µL of Buffer EB. For PCR2, a single primer pair was used to add the remaining Illumina adaptor segments to the ends of the concentrated amplicons of PCR1. The PCR2 (25 µL volume per reaction) consisted of the same combination of reagents that was used in PCR1, along with 5 µL concentrated PCR1 product as template. The PCR 2 conditions were as follows: 2 min denaturation at 98 °C; 12 cycles of 20 s at 98 °C, 30 s at 66 °C and 20 s at 72 °C; and 2 min at 72 °C for final extension. Amplicons were size-selected by gel electrophoresis: gel bands at c. 440bp were extracted and concentrated, using the ZR-96 Zymoclean Gel DNA Recovery Kit (ZYMO Research, Irvine, CA), following manufacturer’s instructions, quantified using a Qubit Fluoromoeter (Invitrogen, NY), and pooled in equimolar concentrations for library preparation for sequencing. Samples were sent to the Georgia Genomics Facility at the University of Georgia (Athens, GA; www.dna.uga.edu), and sequenced on the Illumina MiSeq platform as paired-end reads.

Sequence processing

Raw sequences were processed using the FastX Toolkit (http://hannonlab.cshl.edu/fastx_toolkit) and the QIIME pipeline (Caporaso et al., 2010). Barcodes were recombined from paired-end reads, and forward reads were used for downstream analysis. All sequences were trimmed to 112 bp, including a 12 bp barcode, and low quality sequences were removed. Quality filtering settings were as follows: minimum 30 quality score over at least 75% of the sequence read; no ambiguous bases allowed; 1 primer mismatch allowed. After quality control and barcode assignment, the remaining 1,368,938 sequences were binned into operational taxonomic units (OTUs) at a 97% sequence similarity cutoff using uclust (Edgar, 2010). The highest-quality sequences from each OTU cluster were taxonomically identified using reference sequences from Greengenes (DeSantis et al., 2006). Plant-chloroplast and mitochondrial OTUs were removed. Not all samples returned the same number of sequences. Following rarefaction precedents (e.g., Human Microbiome Project Consortium, 2012; Kuczynski et al., 2010) we rarefied all samples to 500 sequences per sample. Samples with fewer than 500 sequences were not used in subsequent analyses (Table 1); since some low-yield samples were removed from all team groups, players were not considered as paired samples before vs. after a bout. Samples analyzed for this study were compiled from two separate MiSeq runs, and additional aspects of this study were included in the MiSeq runs but are not considered here, so the returned sequence count does not reflect the full volume of the runs. Three of the track samples contained enough sequences to be considered in analysis, and were processed exactly as the rest of the samples, but were not used in any ordination analysis. Sequence files and metadata for all samples used in this study have been deposited in MG-RAST (ID 4506457.3–4506498.3).

Table 1 Description of the two roller derby bouts considered in analyses.

Two different bouts were sampled; bout 2 occurred approximately 5 h after bout 1. Emerald City Roller Girls played in both bouts. Neither team in bout 1 had played a bout previously in the day, but both teams in bout 2 had done so. Total skin samples considered in analysis = 82. Colored points correspond to those used in all figures.

Team	n Players	Bout	1st Bout
of the day	
	Before	After			
Emerald City	 7	 7	1	yes	
Silicon Valley	 10	 4	1	yes	
Emerald City	 14	 14	2	no	
DC	 13	 13	2	no	

Statistical analysis

All statistical analyses were performed in R. Community variation among samples, or β-diversity, was calculated using the quantitative, taxonomy-based Canberra distance, implemented in the vegan package (Oksanen et al., 2011). Non-metric multidimensional scaling (NMDS) was performed using the bestnmds function in the labdsv package (Roberts, 2010), using 20 random starts. Discriminant analysis of within-group similarity was conducted using permutational MANOVA with the adonis function in vegan. To determine whether skin microbial communities became more similar to one another after playing in a bout, we used a β-dispersion test with the betadisper function in vegan. This test is a multivariate analog of Levene’s test for homogeneity of variances (Anderson, Ellingsen & McArdle, 2006), and it tests for a significant difference in sample heterogeneity between groups (i.e. the spread of data points in ordination space). Indicator analysis (Dufrene & Legendre, 1997), using indval in labdsv, was conducted on each team group, and all players combined before vs. after respective bouts, to identify OTUs responsible for observed β-diversity results. P-values for significant indicators were adjusted for multiple comparisons using Holm’s correction (Holm, 1979). The relationship between time played and change in community composition was assessed with Pearson’s correlation test by comparing individual players’ pairwise community distances with their estimated cumulative times during a bout.

Results

Illumina sequencing of the V4 region of the 16S rRNA genes produced 1,368,938 barcoded sequences. After quality filtering and rarefaction, 82 samples were considered during analysis, taken from 3 teams and two bouts (Table 1). Emerald City Roller Girls (EC) played in both bouts and thus were included twice in the analyses as two different team groups; Silicon Valley Roller Girls (SI) and DC Roller Girls (DC) played in the first and second bouts, respectively, against EC. Including EC players twice in the study allowed us to evaluate the change in community composition in a single team after playing successive bouts. Some EC players were sampled in both bouts, but were not analyzed on a paired-sample basis. Rarefaction to 500 sequences per sample left 1034 bacterial OTUs, with the most abundant OTU (Corynebacterium sp.) representing c. 34% of total sequences, and c. 55% of all OTUs represented by singletons (i.e. a single sequence represents a single OTU). The bacterial taxa identified in our skin samples were consistent with what has been reported in other skin microbiome studies (e.g. Caporaso et al., 2011a; Costello et al., 2009; Grice & Segre, 2011; Kong, 2011) When considered at the taxonomic class level, the majority of sequences were Actinobacteria (57.9%), followed by Bacilli (23.4%), Gammaproteobacteria (7.4%), Betaproteobacteria (3.7%), Alphaproteobacteria (2.7%), and Clostridia (1.3%). All skin samples were dominated by skin-associated genera (especially Corynebacterium, Micrococcus, Staphylococcus, and Acinetobacter), and by oral-associated genera (including Neisseria and Rothia), both before and after bouting. Post-bout samples, however, did contain higher relative abundances of a few soil- and plant-associated genera, especially Arthrobacter and Xanthomonas. Normalized OTU richness, at the 500 sequences per sample level (mean richness = 67.6 OTUs), was not significantly different after a bout (t = 0.007; p = 0.9; from a Welch two-sample t-test).

Were players’ skin microbiomes predicted by team membership?

Bacterial communities detected on players’ upper arms from different teams were significantly different before playing a bout, as well as after playing a bout (Table 2). In other words, the skin microbiome of an individual player was predicted by team membership. Teams clustered together in ordination space using a non-metric multidimensional scaling representation of players’ skin microbiomes both before (Fig. 1A) and after (Fig. 1B) playing a bout, based on Canberra taxonomic distances. Though team clustering is significant in both cases (before and after a bout), there is a greater degree of overlap between the teams following bouts. Emerald City was considered as two different teams in two different bouts during analysis.

Figure 1 Variation in skin microbial community composition is significantly explained by team identity.

Ordination diagrams (axes 1 and 2 from separate 3-dimensional NMDS ordinations) summarizing similarity of skin bacterial community composition of all players. (A) Points represent players before bout 1 (EC  vs. SI ) and before bout 2 (EC  vs. DC ). Corresponding-colored ellipses show standard deviations around community variances from each team. The skin bacterial communities of the four team groups were significantly different before playing a bout (p < 0.001; from permutational MANOVA on Canberra taxonomic distances). (B) The four team groups are also significantly different after playing bouts (p < 0.001), though more overlap is observed between teams after bout 1 (EC  vs. SI ) and after bout 2 (EC   vs. DC ). NMDS 3-dimensional stress = 19.66 (A) & 17.55 (B).

Table 2 Results from Permutational MANOVA on Canberra distances among skin bacterial communities sampled from players before and after bouting.

Each team was considered individually when testing for intra-team before/after community shifts, while teams were considered together for the “all players” before/after test. Team identity was used as a grouping factor to test inter-team clustering (“all teams”), both before and after bouts. Emerald City was considered to be two different teams (bout 1 and bout 2) in analyses.

Comparison	Team	DFresid	F-statistic	p-value	Bout	
Before/After	Emerald City	12	1.25	0.017*	1	
	Silicon Valley	12	1.39	0.005*	1	
	Emerald City	26	1.22	0.011*	2	
	DC	24	1.35	<0.001*	2	
	all players	80	1.96	<0.001*	–	
Before	all teams	40	1.74	<0.001*	–	
After	all teams	34	1.27	<0.001*	–	
Notes.

* Significant at p < 0.05 level.

The home team’s pre-bout bacterial communities (EC) were more similar on average to communities detected on the track than the two visiting teams when considered together (p < 0.001; from a Welch two-sample t-test; Fig. 2), and when considered separately (p = 0.001 & 0.007; for bouts 1 & 2, respectively). All players were more dissimilar on average from track samples (mean Canberra distance = 0.89) than from all other players (mean Canberra distance = 0.83; p < 0.001; from a Welch two-sample t-test). Player bacterial communities did not become more similar to the track after a bout, and in fact both bout 2 teams (EC & DC) became less similar to the track following a bout (p = 0.008 & 0.003, respectively; from Welch two-sample t-tests).

Figure 2 Home team (EC) players’ skin microbiomes were more similar to the microbial community detected on the roller derby track than visiting teams.

When each player’s pre-bout skin microbiomes were compared to the microbial communities found on the track surface, Emerald City players’ skin microbiomes were significantly more similar on average to the three track samples than were the skin microbiomes of players from Silicon Valley or DC. The same is true when considering teams on a per-bout basis (p = 0.001 & 0.007; for bouts 1 & 2, respectively).

Were team-specific skin microbiomes different after playing a bout?

When teams were considered separately, bacterial communities detected on players’ upper arms before a bout were significantly different than those detected after the bout in all cases (Table 2; Fig. 3). We also detected a signal of already having played in a bout. Two teams, Emerald City and DC, had already played in a bout the morning of the tournament, and that was a significant predictor of community composition before the second bout (F-statistic = 2.16; p-value <0.001; EC  & DC  in Fig. 1A).

Figure 3 Team-specific micobiomes are significantly different after playing in a bout.

NMDS ordination diagrams summarizing similarity of skin bacterial community composition when all players are compared within their own teams before and after a bout. All ordinations are based on Canberra taxonomic distances. (A) Emerald City before  and after  bout 1; (B) Silicon Valley before  and after  bout 1; (C) Emerald City before  and after  bout 2; (D) DC before  and after  bout 2. Corresponding-colored ellipses are standard deviations on community variances for each group. All teams showed significantly different microbial communities before vs. after a bout. NMDS 3-dimensional stress: A = 8.1, B = 10.47, C = 16.2, D = 17.65.

Did opposing teams’ skin microbiomes become more similar after competing in a bout?

All players’ skin microbiomes were more similar to one another after competing in a bout. To test this we conducted a β-dispersion test, which compared all players before and after bouting. A significant reduction in β-dispersion between groups (before vs. after) confirmed that communities became more similar (based on Canberra distances; F = 11.79; p < 0.001; Table 4; Fig. 4) when all players were considered together, and when players were grouped by the bout in which they played (F = 12.41 & 4.11; p = 0.002 & 0.048; Table 4). A greater proportion of OTUs was shared by competing teams after bouting compared to before (Table 5), but, interestingly, the opposite is true in 3 out of 4 cases when comparing non-competing teams. When each team was considered separately, both teams in bout 1 experienced a significant β-dispersion reduction as did EC after playing in bout 2, while DC did not (Table 4; Fig. 4). None of the four teams experienced a significant shift in Shannon-Wiener OTU diversity or evenness after playing in a bout, nor was there a difference when all players were considered together (all p-values > 0.2). Both teams in bout 2 had already played a bout previously in the day; neither team in bout 1 had played during the same day. Changes in bacterial communities before and after a bout were not correlated with each players time spent in a bout (Pearsons correlation test; ρ = 0.12; p = 0.45). Jammers and blockers play different roles on a team, and thus engage in different amounts of contact and time played; however, since players are not limited to a single position during a bout, we did not differentiate between the positions during analysis.

Figure 4 Bacterial community variance is reduced after playing in a bout for all players and for three of the four teams individually.

When all players were considered, regardless of team identity, bacterial communities were significantly more similar to one another after a bout than they were before a bout (p < 0.001). Both teams in bout 1 (EC and SI), as well as EC in bout 2, showed the same microbial community convergence. Points are jittered around the x-axis to more clearly describe distributions. All p-values are from β-dispersion tests; a lower mean community variance for the “after-bout” points means that players’ skin micobiomes were more similar to one another after playing in a bout. Colored points correspond to Table 1 and Figs. 1 and 3.

Indicator analysis

We conducted indicator analysis to identify OTUs responsible for the observed difference between teams and for the convergence of bacterial communities after playing. Forty-nine OTUs were significant indicators for single pre-bout teams (p-value <0.05) before multiple comparison adjustment, and 11 were significant after adjustment (Table 3); indicator taxa are presented at the finest taxonomic resolution provided by Greengenes assignment. Though the significant indicators include bacterial genera commonly associated with soil and plants (e.g. Dietzia & Xanthomonas), notable human-associated groups also stand out (e.g. Coprococcus, Streptococcus, Lachnospiraceae & Brevibacterium). This list of indicators includes both rare OTUs that account for <1% of sequences, and relatively common OTUs (Alicyclobacillus made up >25% of pre-bout SI sequences). We also identified OTUs that were shared between competing teams in post-bout samples but not in pre-bout samples. Six OTUs fit this description for both bouts, belonging to six different bacterial genera: Streptococcus, Sphingomonas, Eubacterium, Porphyromonas, Aerococcus and Methylobacterium. It is worth noting that although these OTUs showed a shared response after bouting, all six OTUs were relatively rare throughout the study; none accounted for more than 1% of total sequences for any player.

Table 3 Indicator OTUs from each team prior to sampled bouts.

All bacterial OTUs were tested as indicator taxa for any of the four teams before playing in a bout, using Dufrene and Legendre’s (1997) procedure. Only those significant at the p < 0.05 level after multiple comparison adjustments are shown. Percentages are calculated as a share of all sequences in the dataset, and as a share of pre-bout sequences from each team for whom the OTU is indicative. Six of the eight pre-bout indicator OTUs detected in opposing teams’ samples increased ( + ) in mean abundance for opposing team members after playing in a bout.

OTU Genus	Team (a)	Bout	Indicator value	Percent of
total dataset	Percent of
pre-bout
team sequences	Adjusted
p-value	
Dietzia	EC ( + )	1	0.582	0.75	2.06	0.021*	
Coprococcus	EC	1	0.410	0.03	0.31	0.039*	
Alcaligenes	EC	1	0.381	0.01	0.11	0.030*	
Alicyclobacillus	SI (−)	1	0.554	7.21	25.46	0.021*	
Xanthomonas	SI ( + )	1	0.551	0.14	0.84	0.044*	
Alcanivorax	SI	1	0.500	0.03	0.26	0.021*	
Streptococcus	EC ( + )	2	0.552	2.18	3.66	0.021*	
Nesterenkonia	EC (−)	2	0.532	0.39	0.67	0.039*	
Streptococcus	EC ( + )	2	0.499	4.02	4.41	0.030*	
Lachnospiraceae	EC ( + )	2	0.425	0.05	0.11	0.021*	
Brevibacterium	DC ( + )	2	0.680	1.38	2.58	0.021*	
Notes.

* Significant at p < 0.05 level after Holm’s correction for multiple comparisons.

a Indicates whether OTU increased ( + ) or decreased (−) in average abundance when detected in opposing team.

Table 4 Results from β-dispersion ANOVA on Canberra distances when comparing community variances from each team before and after, as well as all players regardless of team identity.

The first four tests describe β-dispersion tests (comparison of within-team bacterial community variance) when each team is considered individually before and after a bout, and the fifth ignores team identity. Results indicate that skin bacterial communities from Emerald City (bout 1) and Silicon Valley players both became more similar following a bout, as did Emerald City from their 2nd bout. But this was not the case for DC after playing in bout 2. Bacterial communities became more similar when players were grouped by the bout in which they played, and when all players were considered in the same analysis.

Team	DFresid	F-statistic	p-value	Bout	
Emerald City	12	11.34	0.006*	1	
Silicon Valley	12	7.16	0.02*	1	
Emerald City	26	6.03	0.02*	2	
DC	24	0.05	0.82	2	
bout 1	26	12.41	0.002*	1	
bout 2	52	4.11	0.048*	2	
all players	80	19.07	<0.001*	–	
Notes.

* Significant at p < 0.05 level.

Table 5 The number of OTUs shared by competing teams increased after a bout.

When the proportion of shared OTUs is compared for each combination of teams, both sets of competing teams saw an increase (↑) after playing in a bout, while 3 out of 4 possible combinations of non-competing teams saw a decrease (↓) in shared OTUs.

	Teams (Bouts)	Percent of OTUs shared	
		Before	After	
Competing teams	EC(1) & SI(1)	28.2	32.7 ↑	
	EC(2) & DC(2)	27.3	29.9 ↑	
Non-Competing Teams	EC(1) & EC(2)	26.5	29.0 ↑	
	EC(1) & DC(2)	28.9	26.7 ↓	
	EC(2) & SI(1)	24.5	21.7 ↓	
	DC(2) & SI(1)	26.5	23.0 ↓	

Discussion

Bacteria are ubiquitous. Those inhabiting the human body have received increased attention in recent years, owing to a greater appreciation of the interrelated nature of humans and their microbiome, an improved understanding of microbial ecology, and an unprecedented ability to detect fine-scale microbial community changes with high-throughput sequencing technology (Human Microbiome Project Consortium, 2012). The skin is the largest organ and an important barrier that regulates microbial entry into the human body. Despite the importance of the skin ecosystem to human health and well-being, we know very little about the forces that shape microbial structure and composition in the skin environment. The present study was designed as a way to understand how human to human contact influences the skin microbiome, since contact has long been acknowledged as a major dispersal vector for skin bacterial communities (Hamburger, 1947; Pittet et al., 2006).

We found that team membership was a strong predictor of skin microbial community composition, and that differences between teams were partly driven by the presence of unique indicator taxa that are commonly associated with human skin, gut, mouth, and respiratory tract. For example, Brevibacterim was the sole indicator taxon for DC. While short reads are limited in their ability to identify bacterial taxa to the species level, this genus contains well known human commensals that are ubiquitous on skin, and have even been studied for their role in foot odor (Dixon, 1996). The strong microbial fingerprint linked to each team could be because they were from three distinctly different geographic locations within the United States (Eugene, OR; San Jose, CA; and Washington, DC), each associated with a different climate, urban setting, and outdoor macrobiota. These cities may also have very different environmental microbiota. Blaser et al. (2012) recently found that human populations from different geographical locations share distinct skin microbial communities. Consistent with the idea that humans carry a microbial fingerprint that reflects where they live, we found that home team (EC) microbiomes were more similar to their home track than either of the visiting teams prior to bouting. This is also the EC practice track, so it is perhaps unsurprising that EC players share some of their microbiome with the track surface since they shed skin cells and frequently come into direct contact with the floor. While a variety of factors likely contribute to this geographic signature, it is plausible that human contact plays a role.

Although each team retained their microbial fingerprint, we found that team microbial communities became more similar to one another after players competed in a bout. This was found when considering all players together, when players grouped by the two different bouts, and when considering each team individually, though the latter was not the case for DC, who played in the second bout. Several reasonable explanations arise given these results: (1) all players were exercising, and exercise produces predictable changes in skin habitat conditions that are likely to affect bacterial communities over time; (2) players were acquiring microbial transients from the built environment; and (3) players were coming into repeated physical contact with their teammates and those from opposing teams, often using the sampled area of their upper arms, and potentially sharing portions of their skin microbiomes. With regards to explanation (1), the current study was not set up to conclusively rule out the potential for exercise-related bodily changes to alter skin bacterial communities. It seems unlikely that 60 min of elevated skin temperature and perspiration would be long enough for microbial growth dynamics to effect the magnitude of changes observed, given that bacterial doubling times generally exceed 20 min even in optimal conditions. It is possible that exercise results in migration from subcutaneous habitats to the skin surface, but little is known about this potential mechanism. Additionally, both bouts resulted in a greater proportion of shared OTUs between competing teams, but not between non-competing teams except when considering only EC over the course of their two bouts, arguing against an overall exercise effect. Finally, none of the teams experienced a shift in Shannon-Wiener diversity or evenness, which would be expected in an exercise-driven community shift, since metabolically active bacteria might come to dominate the community with a change in pH, temperature and moisture at the skin surface.

Explanations (2) and (3) above both derive from dispersal, either from the built environment or from other players. Dispersal from the built environment to skaters is likely, since roller derby and spectator movements stirred up dust from the recreational venue, and players also frequently fall on the floor. Although humans have been estimated to contribute more than 106 airborne microbial cells per-hour (Qian et al., 2012), culture-based disease transmission studies suggest that direct contact with humans and other surfaces is a stronger bacterial dispersal vector than airborne particles (Casewell & Phillips, 1977; Pessoa-Silva et al., 2004; Pittet et al., 2006). We found that human to track surface contact did not seem to explain the observed shifts in community composition, since none of the four team groups became more similar to the track samples after playing in a bout. Given that the proportion of OTUs shared between competing teams increased after both bouts, but not between non-competing teams, human to human contact is the most parsimonious interpretation for the significant changes in skin microbiome we observed. Future research - particularly over longer time scales - is needed to understand the fate of dispersed microbiota and the dynamics of the human skin microbiome.

We know very little about how our social, family, and professional interactions shape our microbial identities. Contact sports are an ideal setting in which to study how human to human interactions influence our microbial ecosystems. As the rise of mega-cities and population growth continues, humans may experience an increased rate of person to person contact mediated by urban living and global travel. To predict the implications of these changes will require, in part, understanding the ecological and evolutionary forces that act on the skin microbiome. A thorough comprehension of the drivers of the skin microbiome is still emerging; novel approaches to studying our skin ecosystems will likely have lasting implications for health care, disease transmission, and our understanding of urban environment microbiology.

Supplemental Information

Supplemental Information 1 Sequencing Adapters

Click here for additional data file.

We would like to thank the players and coaches who facilitated and participated in this study; we are particularly grateful to Burnadeath, Katarina Van Rotten, Rex Havoc, Vexine, Blue Ruin, and Agent Orange. Willamalane Park and Recreation District Human Resources office provided permission to conduct sampling in the recreation facility. H. Arnold, R. Mueller, P. Pillai, J. Reichman, Z. Stephens, A. Womack, M. Naidoo, and Super Cake helped with sampling and development of molecular protocols. We thank members of the Bohannan and Green Labs for valuable input on this research, and also three reviewers for their thoughtful suggestions on the manuscript.

Additional Information and Declarations

Competing Interests

Author Contributions

Human Ethics

DNA Deposition

JL Green is an Academic Editor for PeerJ.

James F. Meadow analyzed the data, wrote the paper.

Ashley C. Bateman analyzed the data.

Keith M. Herkert conceived and designed the experiments, performed the experiments.

Timothy K. O’Connor conceived and designed the experiments, performed the experiments, analyzed the data.

Jessica L. Green conceived and designed the experiments, performed the experiments, contributed reagents/materials/analysis tools.

The following information was supplied relating to ethical approvals (i.e. approving body and any reference numbers):

University of Oregon IRB & Office for Protection of Human Subjects. January 2012.

# 10262011.038

The following information was supplied regarding the deposition of DNA sequences:

MG-RAST # 4506457.3 – 4506498.3

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
