# Peer review of "Significant changes in the skin microbiome mediated by the sport of roller derby"

_PeerJ, doi:10.7717/peerj.53_

## Round 0.1 · original submission · Major Revisions

All three reviewers and I are enthusiastic about the paradigm you’ve established for measuring the potential for transfer of microbes between athletes during sporting events. We all agree that the results are interesting and provocative. The reviewers have made some suggestions that I hope you will be able to accommodate.

The most important point is that we feel you should analyze and present data on the taxa responsible for the patterns found, a point made most emphatically by Noah Fierer. It would be important to know the taxa for which the teams are most distinguished and the taxa that have contributed most to convergence of communities during the bouts. As the reviewers mention, this would help to explain why the teams were different and why they converged. As Catherine Lozupone has pointed out, this will help determine whether the bacteria transferred are a random set (with transfer rates dependent on the abundance of each taxon) or whether some taxa are more easily transferred. I should mention that while PeerJ does not insist that a paper be as exhaustive as it might be for some other journals, I feel that there is no reason not to present critical analyses based on data that you already have.

Also, Reviewer #2 has suggested that you include more details of sampling. I agree, and this is an important point.

I would like to add one more point. Your analysis does not distinguish between migration and successful migration. That is, it is one thing to get there; it is another to get there and then succeed in the new habitat. So, it would have been interesting to test whether the convergences seen during the bouts were successful, in the sense of persisting some time after the bout. In the spirit of PeerJ, I’m not asking you to collect more data for this paper, but I think the issue should be raised in the Discussion.

I hope you’ll be able to address all these points, plus all the other points raised by the reviewers. The one exception is that I don’t think you need to acknowledge the limitation that you have looked at only one sport. I appreciate that this paper is potentially a paradigm former. Please explain carefully how you have accommodated each suggestion.

·

Basic reporting

Basic reporting was fine (more detailed comments provided below in the 'General Comments for the Authors' section)

Experimental design

Experimental design was fine.

Validity of the findings

Findings and conclusions were, in general, supported by the data presented.

Comments for the author

The manuscript describes a fairly straightforward study – the authors compared bacterial communities on the skin of opposing teams before and after a roller derby event. The objectives were to evaluate if team members share similar skin bacterial communities, if contact sports can increase the dispersal of skin bacteria between individuals, and if skin bacterial communities change during the course of a roller derby bout.

In general, I had few issues with this manuscript. The methods were valid, the statistics seemed appropriate, and the results were presented clearly. However, there are critical analyses that are missing from this study. I’ve highlighted these below.

In Figure 1 it is shown that the three teams harbored skin microbial communities that were significantly different from another. However, there is no mention of which taxa were responsible for these differences. This is critical information as it may help determine why the members of the three teams harbored distinct communities. For example, perhaps one team has more dog-derived bacteria from sharing a team vehicle with a St. Bernard or maybe one team has more mouth-derived bacteria from their pre-game rabid frenzy (I’m just guessing here).

Likewise, there is no mention of what taxa drove the observed changes in communities before versus after bouts (Figure 3). After showing that categories of samples harbor significantly distinct communities – the next step should always be to determine which taxa were responsible for the observed patterns. In this case, such information may help ascertain if the shifts were due to increased perspiration, from picking up bacteria off the ground during the bout, or from the post-bout celebratory yogurt shower (who knows? – maybe yogurt is the new champagne).

Without specific information on which taxa were driving the differences between teams or within teams before/after bouts – most of the Discussion is overly speculative. With more data provided on changes in relative abundances, they may be able to rule out some of the hypothesis they presented to explain their results. Information on microbial taxa is always a valuable complement to the beta diversity metrics - the study would be far more powerful by presenting such data and they would have a better chance of identifying the mechanisms underlying the observed patterns.

Reviewer 2 ·

Basic reporting

Acceptable; please see general comments.

Experimental design

Acceptable; please see general comments.

Validity of the findings

Acceptable; please see general comments.

Comments for the author

I enjoyed reading this paper! The questions are important and timely, the study design is clever and unique, and the approaches are state-of-the-art. I believe the MS could be improved by providing more detail and explanation in response to the following questions and comments.

Major comments:

Please describe the inclusion/exclusion criteria. For example, were players excluded who wore protective clothing or equipment covering their upper arms? Same for the sampling timeline – how long before/after bouts began/ended were samples collected on average? Also, who collected the samples – did the players swab themselves, or was one (or more) operator trained to do so? Finally, please describe the upper arm location – outer (dorsal? posterior?) aspect, deltoid area? (It probably has an ‘official’ anatomical descriptor.)

Do blockers and jammers experience varying levels of skin-to-skin contact during jams? (One might expect blockers to have higher levels.) If so, are players’ positions fixed (e.g., Ann is a particularly good jammer; 95% of her playing time is spent in that role)? Perhaps the authors could briefly comment on predicted contact time when describing the two roles (otherwise, why mention them)?

The authors should describe whether the 7 ‘bout 1’ EC players were re-sampled among the 14 ‘bout 2’ EC players, and whether all 14 ‘bout 2’ EC players also played in bout 1. This is relevant to the interpretation of Fig. 1A, which shows that EC before bout 1 looks different than EC before bout 2 (6 hrs later). Viewing Fig. 1A alone, one could say that EC looked less like SV 6 hrs after playing them than just before. Does this imply that post-bout compositional homogenization is temporary? Do players shower and leave the venue between bouts? Or were these two completely separate cohorts of EC players? E.g., different shifts of skaters with different bout histories? The authors suggest this pattern is related to playing an earlier bout (which DC did as well), but this seems speculative in the absence of more info – especially for DC, for whom we have no earlier observations.

The sequencing yield for this platform seems quite low, even if all samples were run on a single lane (the authors imply that several lanes were used). Was the run shared with other studies? The authors should briefly describe whether this was the target yield, or whether this was unexpectedly low. If the latter, one worries about potential bias. Explain, briefly?

The authors favor dispersal dynamics (over exercise-induced selection) as an explanation for the patterns they observe and do a nice job justifying this in the discussion. However, I am curious about the observation (shown in Fig. 1B) that SV and DC appear more similar to each other ‘after bout’ despite having never played head-to-head (as far as we are told). Perhaps the authors would suggest that ‘before’ and ‘after’ bout represent amount of time spent in a common indoor environment favoring dispersal from shared local sources, including from EC (rather than direct team-to-team transfer)? The authors should briefly describe either the history of bouts between SV and DC, or explain why their compositions converged despite having no such history.

In Figure 2, was DC significantly more different from track than was EC? It appears the difference is driven mainly by SV. I would favor displaying means, etc, for each team individually rather than (or in addition to) home and away.

Why were 6 SV players lost to follow-up? Perhaps they didn’t play during the bout (i.e., ‘benched’)? Sore losers? Taken away in ambulance? Please explain.

The authors suggest that ‘oral’-like taxa (Neisseria and Rothia) were present before and after bouts. However, I’m curious whether post-bout homogenization was driven by higher inputs from oral communities? This could decrease dispersion within and between teams b/c the oral microbiome is, on average, less inter-personally variable than skin, according to cited studies. Are key results recovered in the absence of taxa reasonably suspected to originate from the oral cavity?

I like the idea of using sporting tournaments with varying levels of contact between teams from different geographic regions as a model for studying skin microbiome dynamics in the indoor environment! However, in the selection of only 1 sport, we are given a view of only one level of contact, and thus the experimental design is somewhat limited. The authors should briefly acknowledge this limitation. Perhaps a women’s college indoor tennis tournament would complement this work by providing a competitive scenario lacking in direct contact? Or chess – no sweat, no contact. And so forth.

Minor comments:

The response variable(s) should be clear in the abstract. At present, phrases such as ‘shifts in skin microbiome’ are used. Shifts in what, exactly? Please elaborate – shifts in skin microbiome composition, diversity, variability, etc?

Line 27 ‘here we explore how . . . contact . . .' – soften, the study design doesn’t isolate the factor of ‘contact’, it’s confounded by co-localization and shared selective regimes. However, the study questions are nicely stated on lines 38-40.

Line 50 ‘booty’ – after watching on YouTube ‘The Basics of Flat Track Roller Derby’, I see this is standard terminology. However, you might explain for the uninitiated.

Line 120, BLAST --against-- reference sequences from Greengenes?

Line 145, Silicon --Valley-- Roller Girls

Line 212, direct contact --with-- the floor (ouch!)

Finally, this journal discourages anonymous review. While I generally favor openness, I have mixed feelings about this in the review process. For example, I believe anonymity could play a positive role in freeing junior researchers to openly and honestly critique works by senior researchers in their field. But my thoughts on this are evolving. For now, I pledge that I have done my very best to fairly critique this work. I hope you will find this feedback helpful.

All the best,
Diligent but human postdoc reviewer

·

Basic reporting

I believe that this article meets the described standards of basic reporting for publication in PeerJ.

Experimental design

The experiment design is sound and the the hypotheses clearly stated. The Methods are also sufficiently described.

Validity of the findings

Overall, I find no major flaws with the data analysis or the conclusions made. The data have also been made publicly available as required. There was one sentence in the results section that I thought was unclear:

1) Page 7;lines 148-150: “…, with the most abundant OTU (Corynebacterium sp.) representing c. 34% of total sequences and c. 55% of all OTUs represented by a single sequence.” How can a single OTU represent 55% of all OTUs?

Although overall the data analysis was very well done and conclusions supported, I did think that some additional data analyses might strengthen the paper but I don't think would be required to publish.

Specifically, in the discussion on page 9 (line 217) – page 10 about why the communities become more similar across teams after a bout, the authors propose 3 possible explanations: 1) repeated physical contact across players, 2) acquisition of microbial transients from the built environment, or 3) physiological changes induced by exercise. They then argue that the first explanation is more parsimonious than the second because the player bacterial communities from the visiting teams did not become more similar to the track after a bout. While this is true, another observation that they make earlier seems to argue for explanation number 2. Specifically, on Page 7, lines 157-158, they observe that “Post-bout samples, however, did contain higher relative abundances of a few soil- and plant-associated genera, especially Arthrobacter and Xanthomonas.” If not from the track, where are these soil- plant- associated microbes coming from?

Anyway, this got me thinking that it would be cool to look further into what types of microbial taxa are being acquired after a bout and from where. One possible way to go about doing this,would be to first specifically predict which OTUs are newly observed after a bout. Although a new observation could be explained by sampling bias, this could be corrected for by e.g. only looking at OTUs that are detected at high abundance post- but not pre-bout or that are found across multiple team members only after a bout. Then one could ask whether the OTUs with a high probability of having been transferred are 1) found in the opposing team’s pre-bout skin microbiota or 2) are on the track. In cases where neither is the case, one can further assess whether the microbe is usually found e.g. in skin, soil, or in the built environment, for instance by comparing to published datasets of these sample types.

Another interesting question for the transferred microbes whose likely source is the opposing team, would be whether you could have predicted that OTU would be a likely microbe to transfer based on its abundance in the opposing team’s skin. If skin-skin transfers were random, one might expect abundance in the pre-bout skin to be predictive of its likelihood of transfer, but an alternative hypothesis is that some microbial groups might transfer more readily due to e.g. their specific niche/physical location within the skin environment.

Anyway, just some ideas.

---

## Round 0.2 · accepted · Accept

Congratulations. I think you did a great job on this, and I believe this work should inspire other very good work on modes of dispersal of the human microbiome.

·

Basic reporting

good

Experimental design

good

Validity of the findings

good

Comments for the author

I think that the authors addressed the reviewer concerns adequately. Based on the results described on lines 241-244, I think that the next set of experiments should test whether the DC team has stinkier feet than the other teams :-)

·

Basic reporting

OK

Experimental design

OK

Validity of the findings

OK

Comments for the author

I re-read the revised manuscript and the responses to the reviewers. I think the manuscript is much improved. Although the authors were perhaps annoyed by my initial request for the taxonomic analyses, I think the manuscript is much more robust now that those analyses have been included (just my opinion, of course). On an unrelated note, I think the DC roller derby team should be re-named the 'Brave-i-bacteria' in honor of this paper.